# Proapoptotic Index Evaluation of Two Synthetic Peptides Derived from the Coneshell *Californiconus californicus* in Lung Cancer Cell Line H1299

**DOI:** 10.3390/md18010010

**Published:** 2019-12-20

**Authors:** Irasema Oroz-Parra, Carolina Álvarez-Delgado, Karla Cervantes-Luevano, Salvador Dueñas-Espinoza, Alexei F. Licea-Navarro

**Affiliations:** 1Facultad de Ciencias Marinas, Universidad Autónoma de Baja California (UABC), Ensenada 22860, Baja California, Mexico; ioroz@uabc.edu.mx; 2Departamento de Innovación Biomédica, Centro de Investigación y de Educación Superior de Ensenada (CICESE), Ensenada 22860, Baja California, Mexicokcervantes@cicese.mx (K.C.-L.); sduenas@cicese.edu.mx (S.D.-E.)

**Keywords:** lung cancer, apoptosis, conotoxins, apoptosis, Bax/Bcl-2 ratio, caspases, structure alignment

## Abstract

Lung cancer is one of the most common types of cancer, accounting for approximately 15% of all cancer cases worldwide. Apoptosis is the dominant defense mechanism against tumor development. The balance between pro- and antiapoptotic members of the Bcl-2 protein family can determine cellular fate. The venom of predatory marine snails *Conus* is estimated to have 100–400 toxins called conotoxins. The family of α-conotoxins is known to consist of selective antagonists of nicotinic acetylcholine receptors (nAChRs). Lung cancer cells overexpress several subunits of nAChRs and are considered as an excellent target for new anticancer drugs. We compared the cytotoxic effect of two synthetic peptides derived from *Californiconus californicus*, Cal14.1a, and Cal14.1b, which only differ by one amino acid in their sequence, and compared their proapoptotic balance by Bax and Bcl-2 mRNA expression. We determined the caspase-3 and -7 activation to demonstrate apoptosis induction. Results showed that Cal14.1a induces a high Bax/Bcl-2 ratio in H1299 (lung cancer cells). Although Cal14.1b has a cytotoxic effect on H1299 cells, reducing cell viability by 30%, it does not increase the Bax/Bcl-2 ratio, which could be explained by the Glu in the 15th residue, which is crucial for the ability of Cal14.1a to induce apoptosis.

## 1. Introduction

Cancer is a significant public health problem worldwide. Lung cancer is one of the leading causes of cancer-related mortality throughout the world. Approximately 1.61 million new cases are diagnosed each year, with 1.38 million documented deaths, and the five-year survival rate is only 15–17% [1,2,3,4,5]. Even though chemotherapy is the most effective treatment for most malignant tumors [3], a variety of chemotherapeutics can cause severe side effects, such as vomiting, nausea, diarrhea, constipation, oral and gastrointestinal mucositis that lead to anorexia, malabsorption, weight loss, anemia, fatigue, and sepsis [6], leading to painful symptoms that result in the interruption of cancer treatment [3]. In cancer, it is common for neoplastic cells to carry mutations in cell-death-related genes that allow them to survive despite treatment and increase their ability to invade different tissues and organs [7,8]. Apoptosis is the dominant defense mechanism against tumor development, and it is essential for immunity, growth, and tissue homeostasis [9,10]. Dysregulation of apoptosis in cancer cells is considered to be one of the mechanisms of multidrug resistance [11]. The mitochondrial-mediated pathway of apoptosis is regulated by the Bcl-2 family proteins, in which Bcl-2 and Bax are the main antiapoptotic and proapoptotic proteins, respectively [12]. Therefore, the balance between pro- and antiapoptotic members of this family can determine cellular fate. It has been shown that Bax/Bcl-2 ratio may be more important than the expression of either protein alone in determining the execution of apoptosis, since an increased Bax/Bcl-2 ratio upregulates caspase-3, which in turn activates DNAses that execute cell death [13,14,15]. 

Caspases are proteolytic enzymes primarily known for their role in controlling cell death and inflammation. Caspase-3 is an effector enzyme that plays a significant role in apoptosis activation. Increased levels of caspase-3 in tumor cells cause apoptosis and secretion of paracrine factors that promote compensatory proliferation in surrounding healthy tissues [16]. 

Cancer chemoprevention is defined as the use of natural or synthetic compounds to prevent, slow, suppress, or reverse the carcinogenic process [17]. It is estimated that over half the drugs currently used for the treatment of cancer are either natural products or drugs derived from natural products [18]. The uniqueness, chemical diversity, and structural complexity of marine natural products represent an unexploited source of potential compounds for use as biological probes or in drug discovery [19].

Predatory marine cone snails (*Conus*) have attracted the attention of biologists and pharmacologists for the great pharmacological potential of their venom toxins. It is estimated that each *Conus* species produces 100–400 venom toxins called conotoxins with almost no overlap in the toxin repertoire between the 750 species [20,21,22,23,24]. These molecules are well-known for being highly selective to their molecular targets, which include cell membrane receptors or ion channels [25,26,27,28,29]. The largest group of characterized *Conus* spp peptides is the family of α-conotoxins that are selective antagonists of the neuronal muscle subtype nicotinic acetylcholine receptors (nAChRs) [30]. Their disulfide bond frameworks stabilize compact loop structures which are responsible for their high potency, receptor subtype selectivity, and resistance to proteases. Due to their structural stability, relatively small size, and target specificity, conotoxins are regarded as ideal molecular probes for target validation and peptide drug discovery [31].

In 2016, it was reported that a 17 amino acid synthetic peptide derived from *Californiconus californicus* named Cal14.1a has a cytotoxic effect in four lung cancer cell lines (H1299, H1437, H1975, and H661). Apoptosis-related genes were evaluated to validate that Cal14.1a induces apoptosis via caspase-3 and -7 activation. Results showed that Cal14.1a decreased cell viability, activated caspases, and reduced expression of the prosurvival gene *NFKB1* in H1299 and H1437 cell lines [32]. The amino acid residues in the sequence of Cal14.1a and the cysteine pattern are fundamental for the affinity and activity of α-conotoxins. In this study, we evaluate the properties to induce apoptosis of synthetic peptide Cal14.1b; a synthetic peptide also derived from *Californiconus caifornicus* that only differs in one amino acid from Cal14.1a, a Gly residue in position 15 instead of Glu. We compared Bax and Bcl-2 expression through caspase-3 activation and Bax/Bcl-2 ratio in H1299 cells, between Cal14.1a and Cal14.1b treatments in order to demonstrate the importance of amino acid sequence and peptide structure to its function and receptor affinity.

## 2. Results

### 2.1. Cytotoxic Activity of Cal14.1b

The anticancer activity of Cal14.1b was evaluated by colorimetric MTS reactive, an assay commonly used by cancer researchers to assess cell number, cell viability, and cytotoxicity in response to drug treatment that measures mitochondrial enzymatic activity that occurs only in viable, proliferating cells [33]. We can observe a significant reduction of cell viability after 24 h treatment with Cal14.1b, as seen in Figure 1. The cytotoxic effect of Cal14.1a is also shown. Both synthetic peptides can decrease cell proliferation in the H1299 lung cancer cell line. Staurosporine was used as a positive control (5 μM), an indolo(2,3-alpha)carbazole that was discovered in the course of screening extracts of the bacterium *Streptomyces* sp. and has become the lead compound among the family of protein kinase C [34]. C+ showed a significant difference (*p* **< 0.001) compared to C− (nontreated cells) in terms of decreased cell viability. All results were normalized to the negative control (C−, vehicle 1% DMSO).

### 2.2. mRNA Expression of Bax/Bcl-2 in H1299 after Cal14.1a and Cal14.1b Treatment

Using RT-qPCR, mRNA expression of significant genes involved in apoptosis activation and regulation, Bax and Bcl-2 (target genes), were analyzed. Threshold cycles of β-actin (reference gene) and the target genes were determined in each sample. Relative mRNA expression was normalized against C- of each treatment (Cal14.1a, Cal14.1b, C+, and C-) as calculated by the relative standard curve method [32]. Overexpression of antiapoptotic Bcl-2 family proteins facilities tumorigenesis and tumor progression. Apoptotic stimuli, such as DNA damage, activate tumor-suppressor p53, leading to apoptosis via upregulation of proapoptotic genes, including Bax [35]. Even though it has been shown that both Cal14.1a and Cal14.1b decreased cell viability up to 30% on H1299 cells, as seen in Figure 1, here, we observe that Cal14.1a increases the expression of Bax, whereas Cal14.1b has no significant difference compared to C-, as seen in Figure 2. C+ (staurosporine, 5 μM) decreased expression of Bax and increased expression of Bcl-2 compared to C-, contrary to our expectations for being an anticancer compound. Cal14.1a increased the expression of Bax considerably and, to a lesser extent, the expression of Bcl-2. Cal14.1b has no significant difference in Bax and Bcl-2 expression compared to untreated cells. Apoptosis-related genes after peptide treatment had an unexpected result, but an important outcome, and observed a significant difference in the Bax expression between Cal14.1a and Cal14.1b. These data demonstrate that both peptides have differences in apoptosis activation in lung cancer cells, although their sequences only differ by one amino acid.

### 2.3. Bax/Bcl-2 Ratio 

In the present work, we show that 12 h of treatment with Cal14.1a induces higher relative expression of Bax to Bcl-2 at the mRNA level when compared to vehicle-treated cells. In contrast, Cal14.1b induces higher mRNA levels of Bcl-2 related to Bax, as seen in Table 1. These results suggest that Cal14.1a, but not Cal14.1b, induces a proapoptotic state in H1299 cells.

### 2.4. Apoptosis Activation through Caspase-3 and -7

Caspases are a family of cysteine proteases that are key in the execution of cell death programs, such as apoptosis. Proteolytic cleavage leads to essential changes in cell morphology, such as membrane blebbing, DNA fragmentation, and formation of apoptotic vesicles [36]. Effector caspases 3 and 7 are the apoptosis executioners and have several similarities to each other, such as their activities and substrates [37]. Caspase-3 and -7 activation was analyzed using the commercial kit CellEvent™ Caspase-3/7 by fluorescence microscopy. This kit is based on the emission of fluorescence upon cleavage of the fluorescently labeled DEVD peptide. H1299 cells were also stained with Hoechst 33342 and propidium iodide (PI); the latter is used to identify necrotic or apoptotic cells [32]. The membrane-impermeable dye PI binds directly to the DNA, which is only possible upon membrane damage, occurring at late apoptotic or early necroptotic events [38]. 

After incubating H1299 cells for 24 h with Cal14.1a and Cal14.1b (27 μM), we observed the activation of caspase-3 and-7 by Cal14.1a treatment. On the contrary, cells treated with Cal14.1b showed no fluorescence, as well as C- and C+. These results demonstrate that Cal14.1a can activate apoptosis in H1299 cells, as seen in Figure 3.

### 2.5. Cal14.1a and Cal14.1b Structure Prediction

The synthetic conotoxins Cal14.1a and Cal14.1b were modeled using a homology-based prediction program, and their three-dimensional structures were obtained. The different conformation structures are presented in the Appendix A for Cal14.1a, as seen in Appendix A and Cal14.1b, as seen in Appendix A. As depicted in Figure 4, all the refined 3D models of conotoxins had better scores in the Ramachandran plots than the structures obtained by experimental methods. The residue percentage of the most preferred region (A, B, L) for Cal14.1a (91.7%) and Cal14.1b (90.5%) are satisfactory compared with the percentage of the conotoxin MVIIA obtained by NMR spectroscopy (35.0%). In Figure 5, it is evident that, although both conotoxins are related, their 3D structures are different. 

## 3. Discussion

The sequences of the synthetic peptides Cal14.1a and Cal14.1b (GDCPPWCVGARCRAEKC and GDCPPWCVGARCRAGKC, respectively) were based on previous works from our group (both peptides have Cys bond pattern Cys3–Cys12 and Cys7–Cys17). Both peptides are part of a new superfamily named J_2_ [39]. These two synthetic peptides contain four cysteines residues belonging to framework XIV; although this pattern definition is loose, it is defined as having at least one residue separating each of the cysteine residues. Because of this loose definition, structures of framework XIV adopt different structures that are classified into folds F, G, and A2, which can display globular disulfide connectivity [31]. However, the cysteine pattern in Cal14.1a and Cal14.1b, and two prolines followed by first cysteine, are highly conserved with other conotoxins that are active against acetylcholine nicotinic receptors (nAChRs), such as It14a form *Conus literattus*, Pu14a from *Conus pulicarius*, and ts14a from *Conus tessulatus* [40,41,42]. It has been reported that aliphatic residues, such as Leu and Val, that are in critical positions (position 7) in the conotoxin sequence, increase their affinity for nAChRs [42,43]. NAChRs are ligand-ligated ion channels composed of five different subunits, including α, β, δ, ε, or γ, wherein α and β are the main subunits [44,45].

It is widely reported that lung cancer cells express various nAChRs subunits, and their overexpression is involved in disease progression and resistance [46,47,48,49,50]. In a previous work, we reported that H1299 lung cancer cells expressed several nAChRs subunits, especially α5 and α7 [32]. Here, we demonstrate that Cal14.1a and Cal14.1b decrease the cell viability of H1299 cell line by up to 40%, as seen in Figure 1. Therefore, we can assume that both synthetic peptides act against nAChRs expressed on H1299 cells. 

Cancer cells can become resistant to anticancer drugs. One of the mechanisms of anticancer therapy resistance is their capacity to resist drug-induced apoptosis. The Bcl-2 family, including antiapoptotic Bcl-2 and proapoptotic Bax, are the primary regulatory genes of apoptosis [2]. Resistance to apoptosis is known to play a role in drug resistance in non-small cell lung cancer [51]. 

We determined Bax and Bcl-2 mRNA expression on H1299 cells exposed to Cal14.1a and Cal14.1b treatments. The expression pattern had an exciting outcome. Bax mRNA levels increased significantly (15-fold) after Cal14.1a treatment, whereas Cal14.1b showed no significative difference in Bax expression, and C+ decreased its expression compared to the negative control (C-), as seen in Figure 2. We expected the same Bax levels with C+ due to its known apoptosis-inducing activity. Nevertheless, Bcl-2 was significantly increased with Cal14.1a and C+. Surprisingly, it has been reported that increased Bcl-2 expression is associated with some favorable prognostic factors in endometrial carcinoma [13]. For Bcl-2 expression, Cal14.1a again showed no difference between C- levels, as seen in Figure 2. We can assume that Cal14.1b has no implication on Bax or Bcl-2 expression for death cell. Expression of Bax and Bcl-2 proteins may be helpful in predicting clinical outcome, patients survival, and response to chemotherapeutic agents in colorectal carcinoma [52,53,54]. Although several studies have reported the prognostic significance counteracting both Bax and Bcl-2, most of them have failed to find a significant relationship between Bcl-2 expression levels and clinicopathological parameters of colorectal cancer [14].

Kulsoom et al. [11] observed no significant association of Bax or Bcl-2 expression with remission, disease-free survival, or overall survival among acute myeloid leukemia. The expression pattern of Bax and Bcl-2 not only differs between various cancer, but even within the same cancer. This could explain differences in Bax and Bcl-2 expression after Cal14.1a and Cal14.1b treatments. Bax/Bcl-2 ratio can act as a rheostat, which determines cell susceptibility to apoptosis [55]. Lower levels of this ratio may lead to resistance of human cancer cells to apoptosis. Thus, the Bax/Bcl-2 ratio can affect tumor progression and aggressiveness [14]. Our results show that Cal14.1a induces a much higher Bax/Bcl-2 ratio compared to Cal14.1b, as seen in Table 1. It has been reported that Bax and Bcl-2 expression is the most predictive outcome when combined as Bax/Bcl-2 expression ratio in colorectal tumors compared to expression levels of the Bax and Bcl-2 genes alone [14]. A study reported a serine protease obtained from *Nereis virens* (NAP) that increased the Bax/Bcl-2 ratio mRNA expression in H1299 lung cancer cells, which may cause Ca^2+^ overload of the mitochondria and decrease the mitochondrial membrane potential.

Furthermore, this could lead to the release of Cytochrome C and activation of caspases in mitochondria-dependent pathways [1]. It is known that a high Bax/Bcl-2 ratio in laryngeal squamous cell carcinoma is related to favorable prognosis, decreased risk of patient relapse, and higher disease-free survival [56]. Therefore, the high Bax/Bcl-2 ratio appears to be an indicator of good prognostic outcome of cancer. 

We determined caspase-3 and -7 expression after Cal14.1a and Cal14.1b treatments on H1299 lung cancer cells. Increased caspase-3 induces apoptosis [2]. Results have shown that Cal14.1a promotes full activation of caspase-3 and -7 compared to C-, while Cal14.1b and C+ present no activation of caspases, as in C-, as seen in Figure 3. These results could explain the mRNA expression pattern that was observed with both synthetic peptides and higher Bax/Bcl-2 ratio of Cal14.1a. We suggest that Cal14.1b induces cell death, as seen in Figure 1, through an apoptosis-independent pathway [8]. 

Cal14.1a and Cal14.1b only differ in one amino acid, a glutamic acid (Glu) at position 15 instead of glycine (Gly), respectively. It is known that one amino acid substitution has a direct contribution to conotoxin function and affinity for its receptor [57]. Glu is quite frequently involved in proteins’ active or binding sites. This amino acid is charged and polar, and it generally is found on the surface of proteins. Moreover, Gly is unique as it contains hydrogen as its side chain; this means that there is much more conformational flexibility in glycine, and as a result of this, it can reside in parts of protein structure that are forbidden to all other amino acids. If one glycine is changed to any other amino acid, could have a drastic impact on function [58]. A previous study showed that a single amino acid substitution in TxID, an α-conotoxin from *Conus* textile, shifted its selectivity on nAChRs [57]. The structure–activity relationship of RgIA, an α-conotoxin from *Conus regius* using site mutagenesis, revealed that four residues (Asp-5, Pro-6, and Arg-7 at loop I, Arg-9 at loop II) were crucial for its high-affinity binding to the α9β10 nAChR [59]. TxID identified from *Conus textile* is the most potent reported antagonist of α3β4 nAChR, but also exhibits inhibition of α6β4 nAChR. Yu et al. [45] showed that one amino acid substitution of Lys instead of Ser in the original sequence resulted in high selectively for α3β4 and had no affinity for α6β4 nAChR, which suggests that Ser at position 9 plays a vital role in the selectivity of the peptide. 

Hence, the substitution of one amino acid in the conotoxin sequence is crucial for its function and receptor affinity. Here, we demonstrate for the first time that amino acid substitution in Cal14.1a and Cal14.1b causes a significant difference in apoptosis induction in lung cancer cells. 

## 4. Materials and Methods 

### 4.1. Cancer Cells Culture

Human H1299 lung cancer cells (carcinoma, non-small) were cultured with RMPI-16140 medium supplemented with 10% heat-inactivated fetal bovine serum (FBS, Sigma-Aldrich) and 1% antimycotic antibiotic (10,000 units penicillin, 10 mg streptomycin and 25 mg/mL amphotericin B, Sigma-Aldrich). Cells were incubated at 37 °C with 5% CO_2_. The H1299 cell line was acquired from ATCC (Manassas, United States).

### 4.2. Cytotoxicity Assay

Cytotoxicity was evaluated using a colorimetric reaction named CellTiter 96 Aqueous One Solution Proliferation Assay Kit MTS (Promega). In the MTS reaction, a tetrazolium salt is converted into a soluble formazan product (colored) by enzymes in live cells. The amount of formazan is directly proportional to the number of viable cells. A total of 5 × 10^4^ cells per well were added to culture plate. Cells were incubated 24 h at 37 °C with 5% CO_2_. A total of 27 and 28 μM of Cal14.1a and Cal14.1b were added, respectively. As the positive control (C+) staurosporine was used at 5 μM (dissolved in 1% DMSO). For the negative control (C-), cells were treated with 1% DMSO vehicle. After 24 h incubation, 20 μL of MTS were added to each treatment and the number of viable cells were determined by measuring absorbance at 490 nm on microplate reader EPOCH (BioTek, Winooski, United States). 

### 4.3. Quantitative Real-Time PCR (RT-qPCR) Analysis 

RNA extraction and quantitative RT-qPCR analysis was performed according to previous reported method [32]. Briefly, RNA was isolated with Tri Reagent^®^ (Sigma-Aldrich, Toluca, México). RNA was reverse transcribed using a Superscript III cDNA synthesis kit (Invitrogen, Ciudad de México, México). 7500 Real-Time PCR system (Applied Biosystems Ciudad de México, México) and SybrGreen mix was used to obtain PCR reactions with forward and reverse primers designed from the mRNa sequence. mRNA expression was normalized to β-actin expression, and then was compared to C− mRNA expression. Fold changes were expressed as the mean ± SD. 

### 4.4. Fluorescence Microscopy

Caspase-3 and -7 activation were evaluated by fluorescence microscopy using CellEvent™Caspase-3/7 Green Detection Reagent (Life Technologies Ciudad de México, México) as we reported before [32]. H1299 cells were seeded in 96-well plates and incubated 24 h with 27 μM of Cal14.1a and Cal14.1b. Both controls, C+ (5 μM staurosporine) and C−, were also added as treatments. Then, 5 μM of CellEvent™Caspase-3/7 Green Detection Reagent, 10 μg/mL Hoechst 33342 and 50 μg/mL propidium iodide (PI) were added to each treatment and incubated 30 min at 37 °C with 5% CO_2_. Culture plate was exposed to blue channel filter (excitation at 350 nm), green channel (excitation at 502 nm), and red channel (excitation at 535 nm) to capture the fluorescence of CellEvent Caspase-3/7 Reagent, Hoechst 33342 and PI, respectively. Cell imaging was obtained with EVOS FLoid Cell Imaging Station (Life Technologies Ciudad de México, México) at 20× magnification. 

### 4.5. Statistical Analysis 

Data were analyzed by unpaired Student’s *t*-test comparing differences among treatments. GraphPad Prism 7 software was used. A *p* value ≤ 0.05 was considered statistically significant. 

### 4.6. Proapoptotic Index

The proapoptotic index was calculated using the data from relative expression levels of Bax and Bcl-2 mRNA comparted to untreated cells (negative control) and cells treated with staurosporine (positive control). Briefly, the relative expression of Bax and Bcl-2 was calculated in all groups using β-actin as an expression control. Then, the proportion of Bax/Bcl-2 was calculated using this data. If the proportion of Bax/Bcl-2 is higher than 1, then it is considered a proapoptotic signal. Conversely, if it is lower than 1, then the signal is antiapoptotic. Bax/Bcl-2 proportion was calculated from duplicate experiments. 

### 4.7. Homology Modeling 

The three-dimensional structure of each synthetic conotoxin (Cal14.1a and Cal14.1b) was predicted by homology-based modeling using MODELLER v.9.16 [60] through a strategy known as “Advanced Modeling”. BLAST-P was used to identify the potential consensus template structures for modeling. Each synthetic conotoxin was modeled based on three distinct protein scaffolds of the same protein (Omega-conotoxin MVIIA) with 56% identity. The template PDB files were downloaded from the Protein Data Bank (PDB), with PDB ID 1FEO, 1OMG, and 1DW4.

### 4.8. Molecular Dynamics—Simulated Annealing Strategy

After homology modeling, the 3D structures of the synthetic conotoxins were refined by simulated annealing (SA) calculations with software named Nanoscale Molecular Dynamics (NAMD) [61], followed by analysis and visualization of the results using the molecular graphics software Visual Molecular Dynamics (VMD) [62] and MacPyMOL: PyMOL v1.7.4.4 Edu Enhanced for Mac OS X. For quality control purposes, Ramachandran plots of the synthetic conotoxin structures were obtained with PROCHECK [63] and compared to the Omega-conotoxin MVIIA.

The simulations were performed in a box of TIP3P water molecules as the solvent in all cases with periodical boundary conditions, where an NPT ensemble was assumed—constant number of particles (N) and constant isobaric (P) and isothermal (T) conditions. The pressure was set to 1 atmosphere and the temperature to 300 K. These periodical boundary conditions were coupled to annealing (heating) and relaxation (cooling) steps iteratively. After annealing and cooling, each mutated scaffold was subject to molecular dynamics analysis at 300 K and 1 atmosphere for 25 ns. Analysis of atom trajectory coordinates and energies were written to disk every 10 ps. After the simulated annealing and molecular dynamics calculations, the different conformations were grouped based on their total energy stability during the simulation, in order to find the most thermodynamically stable protein conformation by selecting the structure with the longest existence time.

## 5. Conclusions

We demonstrated that Cal14.1a induced apoptosis in lung cancer cell line H1299 via caspase activation and proapoptotic Bax/Bcl-2 ratio. This induction is dependent on a specific amino acid sequence. The position of a Glu at position 15 is necessary for the ability of Cal14.1a to induce apoptosis via Bax/Bcl-2.

## Figures and Tables

**Figure 1 marinedrugs-18-00010-f001:**
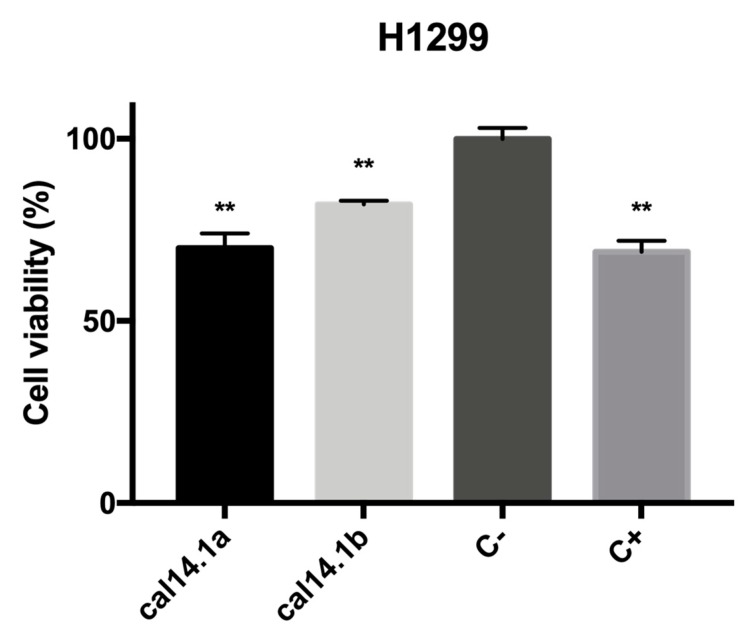
Cytotoxic effect of Cal14.1b. H1299 cells were plated and treated with 27 μM Cal14.1a and Cal14.1b for 24 h. Cell viability was evaluated using MTS assay by measuring absorbance at 490 nm. 5 μM of staurosporine was used as a positive control (C+). Results were expressed as mean ± SEM. *P* ** < 0.01 vs. negative control (C−). Unpaired Student’s *t*-test was used for statistical analysis.

**Figure 2 marinedrugs-18-00010-f002:**
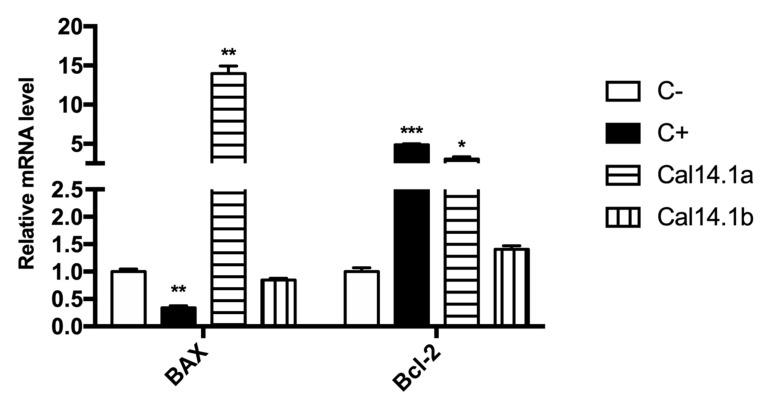
mRNA expression profile of Bax and Bcl-2 in H1299 after Cal14.1a and Cal14.1b treatment. Cells were treated with Cal14.1a and Cal14.1b (both at 56 μM) for 24 h. A total of 2 μg of RNA was reverse-transcribed. mRNA levels were compared by RT-PCR and results were normalized to β-actin gene and expressed as mean ± SD to C-. Staurosporine (5 μM) was used as C+. * *p* < 0.05, ** *p* < 0.01 vs. C- (unpaired Student’s *t*-test).

**Figure 3 marinedrugs-18-00010-f003:**
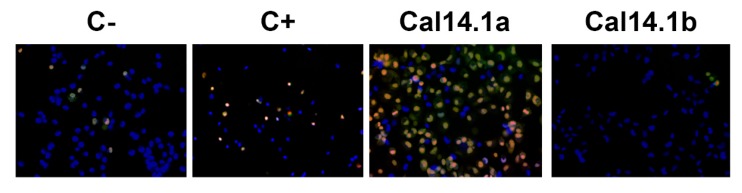
Caspase-3 and -7 activation in H1299 cells after Cal14.1a and Cal14.1b treatment. Image showing C- (vehicle 1% DMSO), C+ (staurosporine 5 μM), Cal14.1a (27 μM) and Cal14-1b (27 μM) 24 h treatments at 460× magnification. Results expressed as positive to caspase-3 and -7 (green) and PI (red).

**Figure 4 marinedrugs-18-00010-f004:**
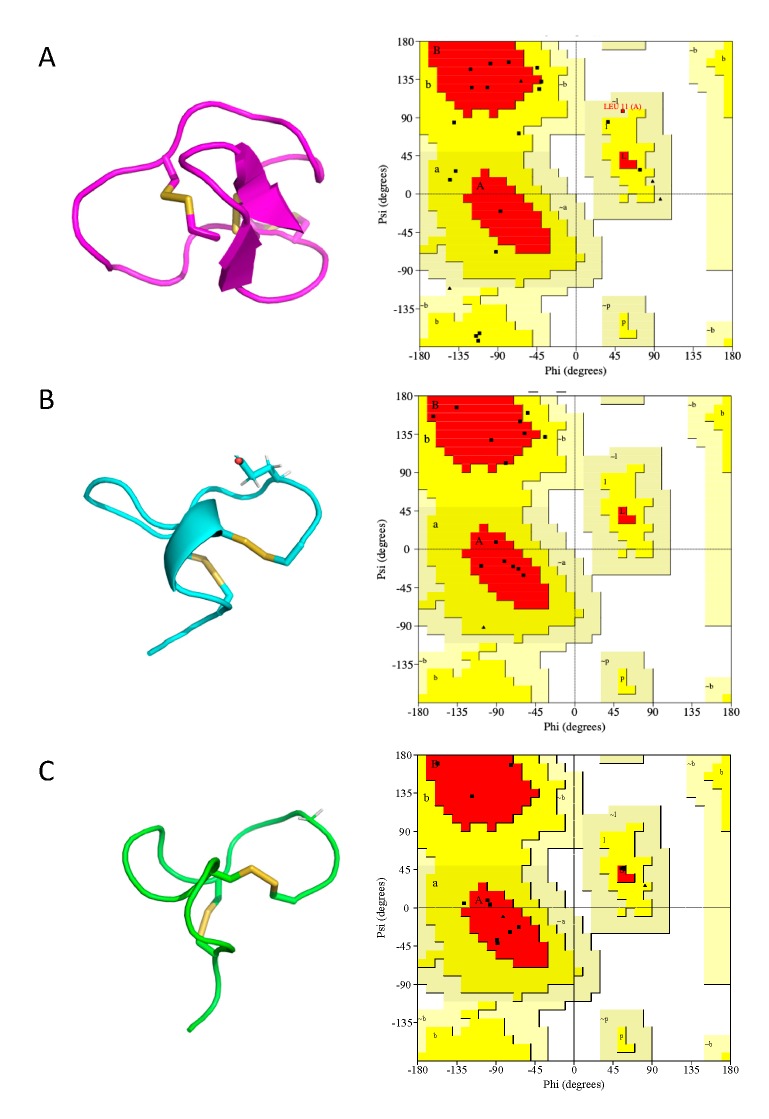
Cal14.1a and Cal14.1b structure prediction compared with conotoxin MVIIA downloaded from www.PDB.org. (**A**) Crystal structure and Ramachandran plot of MVIIA (1FEO) shown in magenta. Refined model structure and their respectively Ramachandran plot of (**B**) conotoxin Cal14.1a shown in cyan and (**C**) Cal14.1b shown in green. The differences of each model are in the amino acid at position 15 (show side chain).

**Figure 5 marinedrugs-18-00010-f005:**
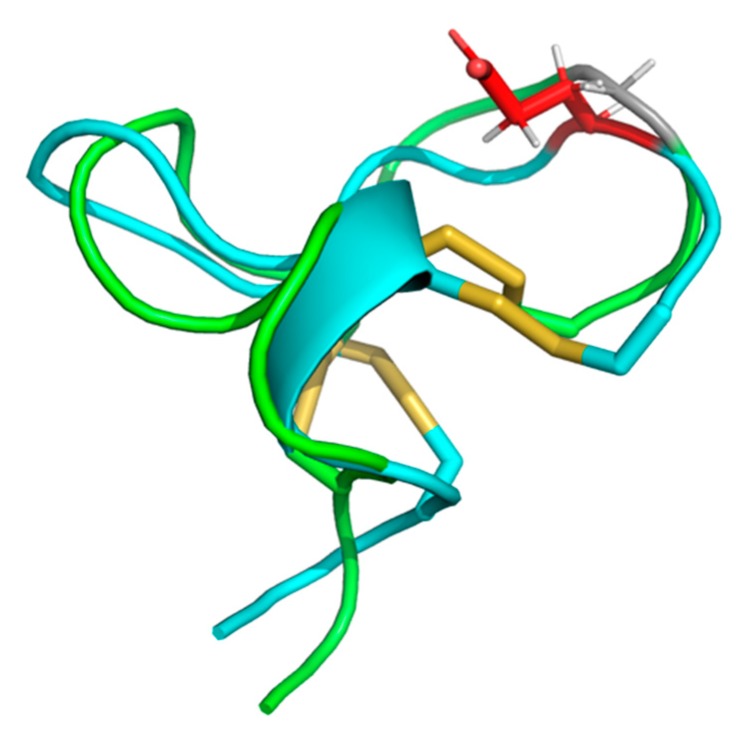
Structural alignment of the models of conotoxins Cal14.1a (cyan) and Cal14.1b (green). The difference between the conotoxins is shown: red—Glu15 in Cal14.1a; gray—Gly15 in Cal14.1b.

**Table 1 marinedrugs-18-00010-t001:** Proapoptotic index (Bax/Bcl-2 mRNA expression).

Peptide	Bax	Bcl-2	Bax/Bcl-2 Ratio
Cal14.1a	5.47	3.06	1.79
Cal14.1b	0.33	1.41	0.24

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
