# Peer review of "Proapoptotic Index Evaluation of Two Synthetic Peptides Derived from the Coneshell Californiconus californicus in Lung Cancer Cell Line H1299"

_marinedrugs, 2019, doi:10.3390/md18010010_

Round 1

Reviewer 1 Report

This manuscript describes the evaluation of two synthetic peptides differing in one amino acid for their cytotoxicity and their capacity to effect apoptosis in a lung cancer cell line. The expression levels of key regulatory genes of apoptosis, which is a defense mechanism against tumor development, the anti-apoptotic Bcl-2 and pro-the apoptotic Bax, were evaluated with quantitative PCR. In addition, the activation of caspase-3 and caspase-7 by the regulator proteins Bcl-2 and Bax was evaluated by fluorescence microscopy. Finally, the results were rationalized by homology modelling and molecular dynamics simulations of these peptides.

The manuscript is very well written, with a good introduction, appropriate experimental design and clearly described methodologies. The results are well presented with the conclusions supported by the results. The described investigation is anticipated to be of considerable interest to readers of the Marine Drugs. However, before publication can be considered this manuscript requires a revision considering the comments listed below.        

Comment 1: The title of the manuscript is very technical. It is not given that the species name ‘Californicus californicus’ is known to a wider potential readership that is interested in conotoxins or coneshells. It is thus suggested to change the title from “Pro-apoptotic index evaluation of two synthetic peptides derived from Californicus californicus in lung cancer cell line H1299” to “Pro-apoptotic index evaluation of two synthetic peptides derived from the coneshell Californicus californicus in lung cancer cell line H1299”.    

Comment 2: The introduction on conotoxins does not do justice to the pioneering work concerning the solid-phase peptide synthesis of these cyclic and disulfide-rich peptides by the group of Paul F. Alewood at the University of Queensland in Australia over several decades. It is thus suggested to include some of the following references in the manuscript: [Accelerated chemical synthesis of peptides and small proteins, Miranda, Les P.; Alewood, Paul F., Proceedings of the National Academy of Sciences of the United States of America (1999), 96(4), 1181-1186] and [Conotoxins: Chemistry and Biology, Jin, Ai-Hua; Muttenthaler, Markus; Dutertre, Sebastien; Himaya, S. W. A.; Kaas, Quentin; Craik, David J.; Lewis, Richard J.; Alewood, Paul F., Chemical Reviews (2019), 119(21), 11510-11549].

Comment 3: The Discussion section on Page 6, Lines 163-164 lists the sequences of the two peptides but is incomplete without the linkage information of the disulfide bridges.      

Comment 4: The Material and Methods section needs to contain a brief description of the chemical synthesis and the purification of the two peptides.  

Comment 5: In the Material and Methods section on the Fluorescence Microscopy on Page 8, Lines 271-272 (“blue channel filter (390-40/446-33 nm), green channel (482-18/532-272 59 nm) and red channel (585-15/646-68 nm”) it is unclear what the notation for wavelength of the fluorescent light is. What is the position of the excitation wavelength and the emission wavelength in this notation and what are the hyphenated numbers after the wavelength?    

Comment 6: In the Material and Methods section on the Molecular dynamics – simulated annealing strategy on Page 9, the number of water molecules in the unit cell needs to be provided, as it gives a sense of the computational ‘expense’ of the calculations.    

Comment 7: The manuscript contains some minor spelling mistakes and grammatical errors, e.g.:

Throughout the manuscript including references: Please italicize the species names.

Page 1, Line 16: Please replace “Bcl-2 family” with “Bcl-2 protein family”.

Page 1, Line 21: Please replace “differs” with “differ”.

Page 5, Line 151: Please replace “NMR” with “NMR spectroscopy”.

Page 6, Line 164: Please replace “was” with “were.

Author Response

We appreciate all the comments of reviewer 1 to improve our work.

Comment 1. The title of the work was changed. 

Comment 2. Suggested references were added.

Comment 3. Information was added.

Comment 4. Chemical synthesis information was added in section 4.1 

Comment 5. Information regarding wavelength was added.

Comment 6. requested information was added: The simulations were performed in a box of TIP3P water molecules as the solvent in all cases.

Comment 7. All minor spelling mistakes and grammatical errors were corrected.

Reviewer 2 Report

Manuscript ID marinedrugs-653051

Pro-apoptotic Index Evaluation of Two Synthetic Peptides Derived from Californicus californicus in Lung Cancer Cell Line H1299.

In this paper the authors present some data on the possible pro-apoptotic activity of two slightly different synthetic peptides derived from venom toxin of marine cone snail.

The two peptides (Cal14.1a and Cal14.1b, that I will call simply A and B) have been used in several experiments in order to demonstrate a different action due to a single aminoacid substitution.  

Peptide A seems to be active in apoptosis induction, evaluating  cytotoxicity, Bax/Bcl2 ratio  and caspase activation. Peptide B shows a moderate effect  only on cell viability.

I have several concerns:

1)  - All the experiments on peptide A has been already published by the same group in 2016, also in that case on the H1299 cell line (see ref. 31, fig. 1, 2 and 3). This fact is not adeguately quoted in the present paper. If we do not consider those results, the outcome of the present paper is quite poor.  

2)  - The authors use a positive control that does not work in two experiments out of three (fig.2 and 3), and, in the case of fig. 3, even in contrast with the results of 2016 !

3) - The authors should schedule some experiments in order to directly prove what stated on line 176 of the text.

I believe that the present paper is below the standard of publication in Marine Drugs.

Author Response

we appreciate all the comments and concerns of the reviewer to improve our manuscript.

concern 1. All the experiments on peptide A has been already published by the same group in 2016.

Although we applied the same methodology as in Oroz-Parra, et al. 2016, in that paper, we only present the cytotoxic effect of synthetic peptide Cal14.1b in four lung cancer cell lines and expression of genes involved in apoptosis Bax, Bcl-2, NFKB-1, and COX-2. Here we compared the pro-apoptotic balance between Bax and Bcl-2 expression in Cal14.1a and Cal14.1b; both synthetic peptides differ only in one amino acid residue in their sequence. In this work, we present the importance of one amino acid for these peptides to lead apoptosis in lung cancer cells. Cal14.1a shows a higher Bax/Bcl-2 ratio in H1299 cell line than Cal14.1b. It has been reported that counteracting Bas and Bcl-2 failed to find a significant relationship between their expression. Therefore, we can demonstrate the most predictive outcome when combined as Bax/Bcl-2 expression ratio compared to expression levels of Bax and Bcl-2 genes alone. The information showed in this work is innovative and relevant compared to our 2016 published paper.

concern 2. The authors use a positive control that does not work in two experiments out of three (fig.2 and 3.

In figure 2, we can appreciate the increase of BCL-2 expression level when the positive control its added. On the other hand, we agree with the reviewer that the positive control in figure 3, did not show the expected result, however, it is clear that Cal14.1a presents a full activation of caspases. 

concern 3. The authors should schedule some experiments in order to directly prove what stated on line 176 of the text.

we agree with the reviewer, we add a sentence concerning this comment.

Round 2

Reviewer 2 Report

Concern 1:

I only partially agree with author's reply, maily because in the present paper you did not clearly quoted the results obtained in 2016 !

Concern 2: I was wrong in indicating fig.2: it is fig.1 the other in which positive control does not show a clear result !

Author Response

Concern 1:

I only partially agree with author's reply, maily because in the present paper you did not clearly quoted the results obtained in 2016 !

Answer: We had more details of our obtained results in 2016; specifically in the Introduction section (lines 73-84).

Concern 2: I was wrong in indicating fig.2: it is fig.1 the other in which positive control does not show a clear result!

Answer: we had a sentence in lines 103-105 were we mention that the positive control has a statistical difference (p=0.001)related to the negative control. We agree with the reviewer that the positive control should be more potent. However, these are the results that we get, and they show a significant difference.

We appreciate all the comments of the reviewers in order to increase the quality of our work.